# HTLM: Hyper-Text Pre-Training and Prompting of Language Models

## Abstract

We introduce HTLM, a hyper-text language model trained on a large-scale web crawl. Modeling hyper-text has a number of advantages: (1) it is easily gathered at scale, (2) it provides rich document-level and end-task-adjacent supervision (e.g. `class` and `id` attributes often encode document category information), and (3) it allows for new structured prompting that follows the established semantics of HTML (e.g. to do zero-shot summarization by infilling `<title>` tags for a webpage that contains the input text). We show that pretraining with a BART-style denoising loss directly on simplified HTML provides highly effective transfer for a wide range of end tasks and supervision levels. HTLM matches or exceeds the performance of comparably sized text-only LMs for zero-shot prompting and fine-tuning for classification benchmarks, while also setting new state-of-the-art performance levels for zero-shot summarization. We also find that hyper-text prompts provide more value to HTLM, in terms of data efficiency, than plain text prompts do for existing LMs, and that HTLM is highly effective at auto-prompting itself, by simply generating the most likely hyper-text formatting for any available training data. We will release all code and models to support future HTLM research.

## 1 Introduction

The vast majority of text used to pretrain language models is extracted from web pages, while discarding any markup they contain Liu et al. (2019); Brown et al. (2020); Raffel et al. (2019); Lewis et al. (2019). We argue that this HTML should not be ignored; it enables new forms of highly effective language model pretraining and prompting with structured document-level supervision.

Hyper-text, such as the HTML found in the Common Crawl[1], has a number of advantages for pretraining over plain text. It often encodes high-level properties of different parts of the documents, which are difficult to infer from the text alone. For example, `<title>` elements can be excellent summaries of the `<body>` of a document, while element `class` and `id` attributes can encode categorical properties of documents. Such supervision is highly diverse, depending on what the website authors choose to present, and provides close proxies for many NLP tasks we aim to later solve.

Modeling hyper-text allows us to introduce *structured prompting* of language models. We design prompts that incorporate the established semantics of HTML to better control for the desired model output. This includes, for example, performing zero-shot summarization by asking the model to infill `<title>` tags in a web page. And, the fact that we jointly model text and hyper-text formatting also allows for effective auto-prompting. If we have even a few examples for a new task, we can directly ask the model to format them in HTML, and templatize the result to define the new prompt.

Our **H**yper**T**ext **L**anguage **M**odel (HTLM) is trained on 23TB of simplified HTML which we automatically extract from common crawl dumps (see Section §2.1). We use a modified BART denoising objective Lewis et al. (2019) that randomly masks spans of hyper-text and aims to reconstruct the original input. We extend the original masking with a new size hint scheme, where each mask is associated with an integer that provides a noisy hint for the size of the masked text, to allow for more fine grained task-specific length priors when prompting the final model (see Section §2.3). Figure 1 shows an example mask that should be reconstructed with a phrase that contains roughly 12 tokens.

---

[1] https://commoncrawl.org/

```
<!DOCTYPE html>
<html>
    <title> <mask>12 </title>
    <body>
        ~ south korea on monday announced
            sweeping tax reforms ,
            including income and corporate
            tax cuts to boost growth by
            stimulating sluggish private
            consumption and business
            investment .
    </body>
</html>
```

```
<!DOCTYPE html>
<html>
    <title> ~ South Korea Announces Tax
        Reforms To Boost Economic Growth ~
        </title>
    <body>
        ~ south korea on monday announced
            sweeping tax reforms...
    </body>
</html>
```

Figure 1: An example structured prompt for a simple summarization task, where we ask a generative masked language model to generate a mask representing the title with an average tokens size of 12.

Through extensive experiments, we show that our `HTLM` achieves highly effective transfer for a wide range of end tasks and supervision levels. It matches or exceeds the performance of comparably sized text-only LMs for zero-shot prompting and full fine-tuning on GLUE, while also setting new state-of-the-art performance levels for zero-shot summarization with a gain of up to 8 ROUGE-1 points. It also allows few shot learning for problems that are less easily reduced to text-only inputs, such table to text generation. Following methodology introduced by Le Scao & Rush (2021), we further find that hyper-text prompts provide more data efficiency to the `HTLM` model than plain text prompts do for existing LMs, being effectively equivalent to having up to a thousand extra training examples. Finally, we see that the `HTLM` model is highly effective at auto-prompting itself, in some cases rivaling the performance of manually engineered prompts.

In summary, our contributions include:

- We present the first hyper-text language model (`HTLM`), trained on 23TB of simplified HTML data from the common crawl.

- Our new hyper-text prompting scheme uses both the well-established semantics of HTML and new size hints on prompt masks to provide more fine-grained control of new task specifications.

- We demonstrate consistently strong transfer from `HTLM` to a range of tasks at differing supervision levels, including improving the best-known zero-shot summarization numbers by up to 8 ROUGE-1 points.

- Following Le Scao & Rush (2021), our data efficiency analysis shows that hyper-text prompts are worth more to the `HTLM` model than plain text prompts are for existing LMs, being effectively equivalent to having up to a thousand extra training examples.

- We demonstrate the `HTLM` directly supports auto prompting for new tasks, by simply asking it to format any available examples in HTML, often rivaling or surpassing previous manually engineered prompts.

- We release all code and models to support future `HTLM` research.

## 2    HYPERTEXT LANGUAGE MODEL (`HTLM`)

`HTLM` is trained on a large corpus of simplified HTML, which is automatically extracted from the common crawl (Section §2.1). We use a BART-style denoising autoencoder with span masking (Section §2.2), extended to allow size hints during reconstruction of the original text (Section §2.3).

### 2.1    MINIMAL HTML

Although HTML contains supervision signals to natural language, the majority of HTML in a modern web page does not provide any significant form of supervision for pretraining. For example, a large portion of a webpage is JavaScript code or CSS, which provides more aesthetics to the page rather than document-level information. Coupling this with the challenges of training transformers on very long sequence lengths (Choromanski et al., 2020; Wang et al., 2020; Beltagy et al., 2020), it was

important to automatically convert web pages to a simplified form, which we call **M**inimal-**HTML** (MHTML), as defined below.

We remove all sub-trees of the HTML DOM[2] which do not contain textual elements of a certain character size (128 for standard textual elements, 64 for lists/tables/spans). We also filter out all *headers*, *footers*, *copyrights*, *forms*, and *iFrames*. We fold consecutive `<div>` elements into a singular `<div>` element with merged attributes. We also remove all attributes which are not `class` or `id` attributes. Lastly, we skip all MHTML documents whose ratio of text to HTML is not greater than $0.46$. Particularly we noticed that MHTML documents whose ratio of text to HTML is low, the average quality of the document tends to be lower as well. These specific cut-offs were found by visually inspecting a set of Common Crawl (CC) documents after application of aforementioned transforms ensuring both a high quality of kept documents while also not filtering too large amount of data. Furthermore we filter out all documents who have a `lang` attribute that is not set to `en`.

Applying these deterministic transformations removes on average 94% of characters from a raw webpage while maintaining the general markup of the document. Furthermore, it allowed close to 85% of MHTML documents to fit into 1024 BPE tokens; the maximum token length for BART and many other existing language models.

One by-product of this type of filtering is that it also produced high-quality documents by default[3]; thus, we opted out of model-based filtering of documents such as CC-100 (Conneau et al., 2019). We used the January 2021 snapshot of Common Crawl, which provided us with 23 Terabytes of MHTML text after filtering.

## 2.2 MODEL

We adopt a BART-style denoising auto-encoder (Lewis et al., 2019) for several reasons. We want to predict arbitrary substrings within the MHTML, conditioned on the rest of the document. This allows us to equally easily (1) use masks during prompting to mark where to generate text associated with model outputs within a web page, and (2) automatically generate prompts by wrapping plain text training examples in masks that allow the model to mark them up by generating MHTML formatting. We also do not know in advance exactly how much text needs to be generated in each case, thereby ruling out the use of more traditional masked language models. We maintain the random masking in the BART objective, not adding anything specific to HTML, as random masking is the super-set of all underlying masking policies.

For all of our experiments, we adopt the same architecture as BART-Large and initialized our models with the BART-Large checkpoint. This model has roughly 400 million parameters.

We trained our augmented BART model for a total of 330,000 steps on 256 GPUs with an effective batch size of 8192. We initialize our model with the original BART-Large model. We train using the Adam optimizer (Kingma & Ba, 2014) and a polynomial decay learning rate scheduler with a peak learning rate of $4e-5$ and $10,000$ warm-up steps.

We do not use the sentence shuffling from the original BART objective, and select a Poisson $\lambda$ of 3.5 for sampling span lengths for masking. We set dropout in the attention to $0.1$ for the first 170k steps, reducing it to $0.0$ thereafter. We also filter out data to only English (`en`) after 170k steps using FastText Joulin et al. (2016). We noticed the perplexity plateaued around 170k steps which is why we simplify the learning process by removing dropout and applying stronger filtering of the English language.

## 2.3 SIZE HINTS

BART allows each mask to be replaced with multiple tokens during the reconstruction. During pre-training, BART masks a span with the length sampled from a Poisson distribution; thus, the model must learn to implicitly predict the length of the masked text. A fundamental problem we

---

[2]The DOM or Document Object Model is an interface that treats an HTML document as a tree structure wherein each node is an object representing a part of the document.

[3]Much of the noise in existing text collections derived from the common crawl comes from artifacts that are introduced when returning the text in the relatively arbitrary order it appeared in the original HTML, before the markup was stripped.

encountered when trying to use standard BART for zero-shot generative prompting is the inability to control the length of the generated text for each mask, even when using various decoding strategies like length penalties.

To allow for more control, we augment BART's masking scheme by introducing size hints. Specifically, we tokenize the noisy estimate of the length of a span directly and insert it right after the span mask token. For example, given the correct mask length $m$, we insert $n$ $\langle mask \rangle$ tokens where $n$ is $\max \left(1, \lfloor \mathcal{N}(m, m * \epsilon) \rfloor \right)$ and $\epsilon$ is a hyperparameter representing how noisy we want these size hints to be. By optionally injecting size hints, we can prompt the model to generate text of roughly some specific length, or by not injecting size hints, we allow the model to model the mask size implicitly. We give size-hints to $80\%$ of masks with the noisiness of size hints $\epsilon = 0.1$.

## 3 HTML-BASED PROMPTING

We use the HTML-based prompting scheme for a range of generation and classification tasks. Broadly, we use HTML templates–either selected manually or generated by the model itself by auto-prompting–to specify the HTML structure of the task. The template is then instantiated with the task input and placeholder mask tokens for the output. The model uses this instantiated template as a prompt. Because BART models reconstruct the full input, we rely on simple heuristics to match the prefix/suffix around any masks and extract the final output.

### 3.1 GENERATION PROMPTING POLICIES

Given that we have optional size hints for masks, a single prompt can generate a wide variety of text; therefore, we discuss multiple policies to select the prompted results. We can decide not to utilize size hints at all and thus remove the need to use any policies, but this comes at the cost of template robustness. Without size hints, a template not only has to express the semantics of the task, but also needs to match the average target length as well; such prompts are brittle and require careful manual design. However, using hints allows us to decouple generation length from the prompt, greatly improving template reuse across related tasks. It is also possible that for a prompt and a specific subset of the data, HTLM will not generate an output from which we can programmatically extract the generated mask; therefore, our policies for size-hints also mitigate this issue.

For every generation task, we first construct a prompt that can generate the correct text semantically, and then we provide size hints equal to the average target of a subset of the training set, $\bar{s}$. If, for a particular input, we are not able to extract a value, we run HTLM on the same prompt, but with our size hint set to $\bar{s} \pm i\epsilon\bar{s}$, from which we select the output with the lowest perplexity, we continue this process at most five times where $i$ represents the current index of the policy. If we still cannot find a valid generated answer, we fall back on the auto-template described in the next section. In experiments, we denote HTLM-Manual-NS (not sized) as our manually engineered prompt with no size hint, while HTLM-Manual-S uses the policy defined here for all generation benchmarks.

### 3.2 AUTO-PROMPTING

To avoid manually engineering prompts, we also explore automatic generation of structured prompts. By training on hypertext, HTLM can learn high-level document semantics that we exploit for prompt creation. We generate prompting templates by asking the model to recover document markups. Specifically, we place $\langle mask \rangle$ tokens around every independent block of data (e.g. summary/article).

We provide an example of auto-prompting for a sample from the Gigaword summarization dataset (Napoles et al., 2012) with the respective masking in Figure 2 . For our generation experiments, we denote HTLM-Auto-NS (not-sized) as the auto-prompt without using size hints, where HTLM-Auto-S uses the size hints based policy described in the previous section.

We found that HTLM auto-prompting was less effective for classification tasks. We hypothesize that this is because generative targets carry significantly more information than a simple binary target token.

```
<mask>
us rejects charges against its ambassador
    in bolivia
<mask>
<mask>
the us state department said wednesday it
    had received no formal word from
    bolivia that it was ...
<mask>
```

```
<html lang="en" xml:lang="en">
    <head>
        <title>
            the us rejects charges against
                its ambassador in bolivia
                | The Washington Post
        </title>
    </head>
    <body>
        <div class = "post-body
            entry-content">
            <p> the us state department
                said wednesday it had
                received no formal word
                from bolivia that it was
                ...
            </p>
        </div>
    </body>
</html>
```

HTLM →

Figure 2: An example of auto-prompting using a sample from the train-set of the Gigaword dataset. HTLM places the summary inside of a `<title>` inside of a `<head>` element, while placing the article in a `<div>` element with an `entry-content` attribute value for attribute `class` which is common on news web-sites.

## 4 ZERO/ONE-SHOT PROMPTING

Perez et al. (2021) argue that zero/few-shot learning cannot happen when prompts are created by tuning on a large amount of development data. To mitigate for this issue all the manual prompts used throughout our experiments are either derived from related papers or developed using a maximum of fifty samples from the train set.

### 4.1 GENERATION

We evaluate HTLM on summarization, a prototypical generation task. For all summarization benchmarks, we use ROUGE-1/2/L as our primary metrics to stay consistent with other literature (Lin, 2004).

Furthermore we benchmark HTLM on a set of three standard natural language generation tasks. We utilize the official benchmarking scripts provided which report BLEU (Papineni et al., 2002), NIST (Belz & Reiter, 2006), METEOR (Lavie & Agarwal, 2007), ROUGE-L (Lin, 2004), CIDEr (Vedantam et al., 2015) and TER (Snover et al., 2005). We use Li & Liang (2021) for our baselines, and present prefix tuning results with 0.1% of parameters as well.

**Gigaword** consists of headlines from news articles (Napoles et al., 2012). The target summaries are relatively short, consisting roughly on average of 10 BPE tokens.

**CNN/Dailymail** (Hermann et al., 2015) provides multi-sentence target summaries close to 3 sentences, or roughly 50 tokens.

**Reddit TIFU** (Kim et al., 2018) contains summaries of Reddit posts. Specifically, we use the *short* subset of data . Compared to our other summarization datasets, this dataset is highly abstractive and not based on news articles.

**XSum** (Narayan et al., 2018) provides abstractive single sentence summaries of news articles.

**E2E** Novikova et al. (2017) is a table-to-text generation dataset containing approximately 50K samples with 8 unique fields from the restaurants domain.

**WebNLG** Gardent et al. (2017) is also a structured generation dataset containing 15 different domains from DBPedia. We report numbers on the Seen (S), Unseen (U) and All (A) subsets of the data.

**DART** Nan et al. (2020) is a open-domain structured generation dataset containing Wikipedia tables.

We manually searched for prompts for each of these datasets using a maximum of 50 data points from the train set to evaluate the prompts. We compare 3 different types of prompts; HTLM-Manual denotes manually engineered prompts with size hints, while HTLM-Auto-S and HTLM-Auto-NS indicate autoprompting with and without size hints respectively. For our baseline, we compare against

PEGASUS (Zhang et al., 2019), the current state of the art for zero shot summarization. PEGASUS was explicitly pre-trained for summarization by masking and generating salient *gap* sentences from news articles. We present our results in Table 1.

| Model | Gigaword | CNN/DM | Reddit TIFU | XSum |
|---|---|---|---|---|
| PEGASUS-0S | 23.39/07.59/20.20 | 32.90/13.28/29.38 | 14.66/3.06/10.17 | 19.27/3.00/12.72 |
| HTLM-Auto-NS | 27.56/10.17/24.57 | 33.40/13.45/30.10 | 6.71/1.98/7.86 | 15.15/2.54/10.91 |
| HTLM-Auto-S | 28.73/11.31/26.49 | 34.65/14.54/32.15 | 8.15/2.92/9.75 | 17.14/3.41/13.43 |
| HTLM-Manual | **31.61/10.80/28.60** | **38.51/16.10/33.89** | **15.81/2.98/10.54** | **22.34/4.12/14.56** |

Table 1: HTLM results on zero-shot summarization. HTLM-Manual denotes manually engineered prompts with size hints, while HTLM-Auto-S and HTLM-Auto-NS indicate autoprompting with and without size hints respectively. Metrics shown are ROUGE-1/ROUGE-2/ROUGE-L respectively.

| | E2E | | | | | WebNLG | | | | | | | | | DART | | | | | |
|---|---|---|---|---|---|---|---|---|---|---|---|---|---|---|---|---|---|---|---|---|
| | BLEU | NIST | MET | R-L | CIDEr | BLEU S | BLEU U | BLEU A | MET S | MET U | MET A | TER↓ S | TER↓ U | TER↓ A | BLEU | MET | TER↓ | Mover | BERT | BLEURT |
| **Fine-tuning** | | | | | | | | | | | | | | | | | | | | |
| GPT-2$_{MEDIUM}$ | 68.2 | 8.62 | 46.2 | **71.0** | 2.47 | 64.2 | 27.7 | 46.5 | 0.45 | 0.30 | 0.38 | **0.33** | 0.76 | 0.53 | 46.2 | **0.39** | 0.46 | **0.50** | **0.94** | 0.39 |
| GPT-2$_{LARGE}$ | 68.5 | 8.78 | 46.0 | 69.9 | 2.45 | 65.3 | 43.1 | 55.5 | **0.46** | 0.38 | **0.42** | **0.33** | 0.53 | 0.42 | 47.0 | **0.39** | 0.46 | **0.51** | **0.94** | **0.40** |
| HTLM | **70.3** | **8.90** | **46.3** | 70.8 | **2.47** | **65.4** | **48.4** | **55.6** | **0.46** | **0.39** | **0.42** | **0.33** | 0.51 | 0.40 | **47.2** | 0.39 | **0.44** | 0.51 | **0.94** | **0.40** |
| **Prefix (0.1%)** | | | | | | | | | | | | | | | | | | | | |
| GPT-2$_{MEDIUM}$ | 69.7 | 8.81 | 46.1 | 71.4 | 2.49 | 62.9 | 45.6 | 55.1 | 0.44 | 0.38 | 0.41 | 0.35 | 0.49 | 0.41 | 46.4 | 0.38 | 0.46 | **0.50** | **0.94** | 0.39 |
| GPT-2$_{LARGE}$ | **70.3** | **8.85** | **46.2** | **71.7** | **2.47** | 63.4 | 47.7 | **56.3** | 0.45 | **0.39** | **0.42** | 0.34 | 0.48 | 0.40 | 46.7 | **0.39** | 0.45 | **0.51** | **0.94** | **0.40** |
| HTLM | 70.1 | **8.85** | 46.1 | 71.2 | 2.45 | **64.8** | 46.1 | **56.3** | **0.46** | 0.38 | **0.42** | **0.33** | 0.47 | 0.40 | **47.1** | **0.39** | 0.45 | 0.50 | **0.94** | 0.39 |
| **One-Shot** | | | | | | | | | | | | | | | | | | | | |
| HTLM | 32.1 | 3.35 | 24.1 | 31.6 | 0.78 | 28.1 | 18.5 | 22.8 | 0.24 | 0.21 | 0.12 | 0.78 | 0.79 | 0.78 | 22.1 | 0.12 | 0.91 | 0.25 | 0.78 | 0.22 |
| **Base-lines** | | | | | | | | | | | | | | | | | | | | |
| TILB-Pipeline | - | - | - | - | - | 44.34 | 20.65 | 35.29 | 0.38 | 0.21 | 0.30 | 0.48 | 0.64 | 0.56 | - | - | - | - | - | - |
| UIT-VNU-Pipeline | - | - | - | - | - | 19.87 | 0.11 | 7.07 | 0.15 | 0.03 | 0.09 | 0.78 | 0.87 | 0.82 | - | - | - | - | - | - |

Table 2: We evaluate GPT-2$_{MEDIUM}$, GPT-2$_{LARGE}$ and HTLM on table-to-text generation on E2E (left), WebNLG (middle) and DART (right).

HTLM with manual prompts (HTLM-Manual) and size hints substantially improves over state-of-the-art zero-shot summarization results on all four datasets without any tailored pretraining. In particular, we see large improvements of more than 8 ROUGE-L F1 for the Gigaword dataset. Furthermore, size hints-based auto-prompting (HTLM-Auto-S) outperforms PEGASUS in three out of four datasets. Specifically, for the Gigaword dataset, we outperform previous state-of-the-art zero-shot results from PEGASUS by roughly 6 ROUGE-L points. HTLM improvements stem from the fact that HTML-based prompting allows us better control over dataset-specific attributes such as length and style.

For NLG tasks, we required the use of a single training example to get prompting to work sufficiently. We report these one-shot numbers in Table 2. Because these tasks require structured tabular inputs, it is not obvious how to prompt any other text-based pre-trained models. We report other non-trainable baselines such as the grammar based pipeline approaches (TILB/UIT-VNU) in Gardent et al. (2017). To the best of our knowledge, these are the first one-shot table to text, natural language generation results.

## 4.2 CLASSIFICATION

For prompting in the classification setting, we select 4 datasets to work with. Instead of relying on generative prompting to generate target token(s) denoting the correct class, we instead rely on perplexity measures over the set of all targets to select the correct class. In other words, we select the class for which the perplexity of the corresponding instantiated template is the smallest.

**RTE** (Bentivogli et al., 2009) is a textual entailment task formulated as binary classification. We place the candidate in a `<div>` element with the class attribute set to *candidate* and do the same with the respective hypothesis. In the third element, we utilize the prompt from Brown et al. (2020) with the class attribute set to *answer*.

**BoolQ** (Clark et al., 2019) is a yes/no question answering task, also formulated as binary classification for question, passage, and answer triplets. We represent the question as a `<div>` element with the

itemprop set to *https://schema.org/Question*, passage as a *div* element with class attribute *passage* and answer as a *div* element with the itemprop set to *https://schema.org/Answer*.

**Winogrande** (Levesque et al., 2012) consists of adversarially collected Winograd Schema Challenge Levesque et al. (2011) data. We utilize the same template as GPT-3 but place it in a QA style template similar to BoolQ. Please refer to the Appendix for exact templates.

**HellaSwag** The last dataset we evaluate is the commonsense natural language inference task HellaSwag which, due to its adversarial nature, is considered complex (Zellers et al., 2019).

We present our results on zero-shot classification in Table 3. `HTLM` prompting of classification datasets outperforms the most comparable (in terms of number of parameters) GPT-3 Medium sized model on the majority of tasks, while approaching—and on RTE outperforming—the GPT-3 Large model which consists of roughly double the amount of parameters as `HTLM`.

|  | RTE | BoolQ | Winogrande | HellaSwag | # Params |
|---|---|---|---|---|---|
| GPT-3 | 63.5 | 60.5 | 70.5 | 78.9 | 175B |
| GPT-3 Large | 48.4 | 58.9 | 57.4 | 51.0 | 760M |
| GPT-3 Med | 49.8 | 60.3 | 52.1 | 43.6 | 350M |
| `HTLM`-Manual | 51.2 | 55.3 | 54.8 | 47.9 | 400M |

Table 3: Classification accuracy with zero shot prompting. We compare our performance to the full GPT-3 model as well as variants of comparable size.

## 5 FINE-TUNING EXPERIMENTS

In addition to our previous prompting results, we also aim to show that `HTLM` learned representations are useful in the full finetuning setting at least as well as other pre-trained representations. We compare against other pre-training MLM models such as RoBERTa Liu et al. (2019), original BART Lewis et al. (2019), and T5 Raffel et al. (2019) by finetuning on the GLUE benchmark (Wang et al., 2018).

During finetuning, instead of a simple concatenation of sentences from the train set, we place the examples into prompts derived from Le Scao & Rush (2021). We defer to the Appendix for the exact prompts. Given the recent advancements in finetuning, we also report results using the recently proposed R3F method for finetuning (Aghajanyan et al., 2020a) for both RoBERTa and `HTLM`. We note that there are other numerous pre-training methods we do not compare against (such as Devlin et al. (2018); Clark et al. (2020); He et al. (2020); Meng et al. (2021)) but the goal of our fine-tuning results is to show that there is no significant degradation in quality from training on structured documents.

|  | MNLI Acc-m/mm | QQP Acc | RTE Acc | QNLI Acc | MRPC Acc | CoLA Mcc | SST-2 Acc | # Params |
|---|---|---|---|---|---|---|---|---|
| RoBERTA | 90.2/- | 92.2 | 86.6 | 94.7 | 89.1 | 68.0 | 96.4 | 330M |
| RoBERTa-R3F | 91.1/91.3 | 92.4 | 88.5 | 95.3 | 91.6 | **71.2** | 97.0 | 330M |
| T5-Base | 87.1/86.2 | 89.4 | 80.1 | 93.7 | 87.5 | 51.1 | 95.2 | 220M |
| T5-Large | 89.9/89.6 | 89.9 | 87.2 | 94.8 | 89.9 | 61.2 | 96.3 | 770M |
| BART-Large | 89.9/90.1 | 92.5 | 87.0 | 94.9 | 90.4 | 62.8 | 96.6 | 400M |
| `HTLM` | 90.3/91.4 | 92.6 | 87.1 | 95.1 | 90.8 | 64.3 | 96.9 | 400M |
| `HTLM`-R3F | 91.4/92.1 | 92.8 | 89.1 | 95.4 | 91.5 | 69.4 | 97.1 | 400M |
| `HTLM`-R3F-Prompt | **91.6/91.2** | **92.9** | **89.4** | **95.7** | **91.7** | 69.8 | **97.3** | 400M |

Table 4: Results on the GLUE development set for various fine-tuning methods applied to `HTLM`.

We present our results in Table 4. Overall `HTLM` is competitive with other pre-training methods. We also note that we can improve fine-tuning performance by placing the examples into prompts and fine-tuning the classification head. The improvements that we see in terms of prompting have no

adverse effects on fine-tuning but are rather positive, providing further evidence that the proposed approach of structured pre-training is a viable alternative to other methods of pre-training even for fine-tuning.

We also show our fine-tuning results for the table-to-text generation datasets in Table 2. Similar to GLUE fine-tuning, we place all NLG samples into a prompt while fine-tuning. HTLM fine-tuned is able to outperform both variants of the GPT-2 model consistently.

## 6 PROMPT DATA EFFICIENCY

| | Average Advantage (# Training Points, P vs. H) | | | | |
| | MNLI | BoolQ | CB | RTE | WiC |
| --- | --- | --- | --- | --- | --- |
| RoBERTa-Large | $3506 \pm 536$ | $752 \pm 46$ | $90 \pm 2$ | $282 \pm 34$ | $-424 \pm 74$ |
| T5-Large | $5010 \pm 230$ | $650 \pm 85$ | $150 \pm 8$ | $300 \pm 65$ | $-220 \pm 20$ |
| BART-Large | $4020 \pm 220$ | $450 \pm 55$ | $125 \pm 10$ | $305 \pm 25$ | $-110 \pm 45$ |
| HTLM | $\mathbf{6025 \pm 440}$ | $\mathbf{855 \pm 205}$ | $\mathbf{255 \pm 35}$ | $\mathbf{840 \pm 45}$ | $\mathbf{45 \pm 25}$ |

Table 5: Average advantage (higher is better) in terms of training points for fine-tuning well-structured prompt ($P$) against a classical classification head ($H$).

| | Average Advantage (# Training Points, P vs. N) | | | | |
| | MNLI | BoolQ | CB | RTE | WiC |
| --- | --- | --- | --- | --- | --- |
| RoBERTa-Large | $150 \pm 252$ | $299 \pm 81$ | $78 \pm 2$ | $404 \pm 68$ | $-354 \pm 166$ |
| T5-Large | $300 \pm 120$ | $350 \pm 95$ | $150 \pm 4$ | $608 \pm 90$ | $20 \pm 43$ |
| BART-Large | $200 \pm 180$ | $325 \pm 54$ | $85 \pm 8$ | $512 \pm 64$ | $-80 \pm 89$ |
| HTLM | $\mathbf{692 \pm 240}$ | $\mathbf{565 \pm 143}$ | $\mathbf{255 \pm 34}$ | $\mathbf{640 \pm 45}$ | $\mathbf{80 \pm 40}$ |

Table 6: Average advantage (higher is better) in terms of training points for fine-tuning well-structured prompt ($P$) against a prompt with a non-sensical verbalizer ($N$).

What does the HTML-based pretraining and prompting scheme offer over one based on the plain text? Le Scao & Rush (2021) explored quantifying how many data points a single prompt was worth. Specifically, they analyzed three different task-specific settings given a pattern (the structure that the inputs are put into) and verbalizer (i.e., yes/no answer to pattern): (1) fine-tuning a classification head ($H$), (2) fine-tuning the verbalizer of a prompt encoding the semantics of the task ($P$), and (3) fine-tuning the prompt but with a verbalizer that is non-sensical ($N$).

By carefully selecting the number of data points to be used during training in each setting while matching the end fine-tuning performance, we can empirically measure the efficacy of prompts in terms of data points. We provide the same analysis extended to BART, T5-Large, and HTLM using the same PET prompts provided in Schick & Schütze (2020). For HTLM, we wrap all PET prompts in an HTML element. We select the same datasets that were used in the original paper for our experimentation; MNLI (Williams et al., 2018), BoolQ (Clark et al., 2019), CB De Marneffe et al. (2019), RTE (Bentivogli et al., 2009), WiC Pilehvar & Camacho-Collados (2019).

We first look at the average advantage of fine-tuning a prompt ($P$) against a classification head ($H$) in Table 5. We see that across the board, HTLM prompts—i.e., hypertext prompts applied to HTLM—are worth more than natural language prompts to various other pre-trained models. Compared to RoBERTa-Large on smaller datasets, HTLM's advantage is close to triple on CB and double on RTE. Furthermore, on WiC, HTLM is the only pre-trained model capable of having a positive training advantage when using prompts. We view this as additional evidence to the benefit of pre-training on structured data on the prompting of a pre-trained model.

We also compare the average advantage of fine-tuning a prompt with a verbalizer ($P$) that makes sense against against finetuning a prompt where we change the verbalizer to a random first name ($N$). This is important to capture whether the benefits arise from representing the data in their respective

patterns or the coupling of the pattern and the verbalizer. We present our results in Table 6. Relative to the previous $P$ vs. $H$ setting we lose a large amount of advantage, as was similarly seen in (Le Scao & Rush, 2021). Interestingly enough for small datasets such as CB, all of the training advantage of the prompt comes from the pattern in `HTLM`.

We view this as further evidence that a structured, document level approach to both pre-training and prompting can be seen as a viable alternative to a purely natural language approach.

## 7 RELATED WORK

GPT-2 Radford et al. (2019) showed that large language models show varying levels of zero-shot performance across NLP tasks when compared to supervised baselines (e.g., rudimentary performance on summarization, but more competitive results on reading comprehension). Brown et al. (2020) through their GPT3 model showed that by further scaling up language models on a large subset of the internet, prompting could be a viable alternative to standard finetuning. The success of GPT3 was largely attributed to massive size and compute-intensive pretraining. By reformulating NLP tasks as cloze-style questions, Schick & Schütze (2020) shows that the prompting capabilities exhibited by GPT3 can occur in language models of a much smaller scale when gradient-based finetuning is combined with task-specific unlabeled data. Follow-up work Tam et al. (2021) improves upon these results without depending on unlabeled data. Unlike GPT-3 and other models which use conventional natural language text-based prompting, we focus on a new hyper-text based prompting scheme using generative masked language models pre-trained directly over HTML.

For task-specific zero-shot performance, custom pre-training and data augmentation schemes have been developed. For example, PEGASUS (Zhang et al., 2019) proposes a novel pre-training scheme tailored for summarization which involves masking and generating salient *gap* sentences from a large news corpus. While PEGASUS is capable of doing zero-shot summarization, it offers little control over summary attributes such as length and style which vary across different summarization datasets. WikiTransfer Fabbri et al. (2021) fine-tunes pretrained models on pseudo-summaries, produced from generic Wikipedia data, which contain characteristics of the target dataset, such as the length and level of abstraction. Our proposed model allows fine-grained control over the length of the generated text by specifying the size of the mask. Moreover, by using different prompts, `HTLM` can produce stylistically varied summaries without dataset-specific augmentation and finetuning.

Another line of work has been looking at a hybrid form of prompting that attempts to optimize very few parameters to solve an end task. For example Li & Liang (2021) argue that optimizing in the continuous prompt space is an effective solution to prompt search while Aghajanyan et al. (2020b) optimize for a low-rank projection of the full parameter space. For simplicity, we only focus on either full-finetuning or zero-shot prompting in this paper.

Attempts have been made to encode architectural priors for structured inputs into transformers as well. Specifically, Ainslie et al. (2020) discuss a new type of model which allows for scalability in input length as well as the ability to encode the structure of the input. We opt to allow `HTLM` to learn the structure that is available in the HTML directly without encoding any structural priors into the model itself.

## 8 CONCLUSION

In this paper, we proposed `HTLM`, a hyper-text language model trained on simplified HTML documents from a large-scale web crawl. We showed that by directly modeling HTML through a BART-like objective, we could do structured zero-shot prompting by representing tasks in HTML. Specifically, we outperform the previous best results on zero-shot prompting for summarization by a wide margin by creating prompts that capture the underlying semantics of each summarization dataset. Furthermore, we show that pre-training on structured data improved full finetuning performance relative to other pre-trained models that only modeled natural language.

We also showed additional advantages of modeling hyper-text, beyond improved accuracy. `HTLM` can be used for auto-prompt by simply asking the model to recover the document structure from training samples; these auto-prompts on datasets like Gigaword and CNN/DM outperformed previous state-of-the-art zero-shot approaches. Lastly, we provided an in-depth comparison of the training

advantage, in terms of data efficiency, that `HTLM` had compared to other pre-training approaches. Across the board, HTML prompts were worth more to `HTLM` than natural language prompts were worth to our baselines, further showing the efficacy of pre-training structured data.

Future work can focus on the scaling laws of structured pre-training and prompting. As was seen from GPT-3, the size of the model and the amount of compute utilized and significant impact on prompting performance.

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

## A APPENDIX

### A.1 FINETUNING HYPER-PARAMETERS

For our GLUE related experiments the following parameters are used.

| Hyper Parameter | MNLI | QNLI | QQP | SST-2 | RTE | MRPC | CoLA |
|---|---|---|---|---|---|---|---|
| Learning Rate | 5e-6 | 5e-6 | 5e-6 | 5e-6 | 1e-5 | 1e-5 | 1e-5 |
| Max Updates | 123873 | 33112 | 113272 | 20935 | 3120 | 2296 | 5336 |
| Max Sentences | 8 | 8 | 32 | 32 | 8 | 16 | 16 |

Table 7: Task specific hyper parameters for GLUE experiments

| Hyper parameter | Value |
|---|---|
| Optimizer | Adam |
| Adam-betas | (0.9, 0.98) |
| Adam-eps | 1e-6 |
| LR Scheduler | polynomial decay |
| Dropout | 0.1 |
| Weight Decay | 0.01 |
| Warmup Updates | 0.06 * max updates |

| Hyper parameter | Value |
|---|---|
| $\lambda$ | [0.1, 0.5, 1.0, 5.0] |
| Noise Types | $[\mathcal{U}, \mathcal{N}]$ |
| $\sigma$ | $1e-5$ |

Table 8: Hyper parameters for R3F experiments on GLUE

