# OpenReview forum: "HTLM: Hyper-Text Pre-Training and Prompting of Language Models"
_ICLR.cc/2022/Conference — ICLR 2022 Poster_

### Official Review · Reviewer_36N9 · 2021-10-29

**Correctness:** 3
**Technical Novelty And Significance:** 2
**Empirical Novelty And Significance:** 3
**Recommendation:** 6
**Confidence:** 4

**Main Review:**

## Strength
1. the writing is mostly linear and easy to follow if the reader is familiar with the field, especially, BART (see additional comments below on writing).
2. the authors are transparent about how to replication process (data preprocessing, training tricks, hyper parameters, etc.) and the authors pledge to release their work.
3. the proposed model seems to be a strong zero/one-shot learner (both text generation, and classification) and appears to be on par with strong existing baseline.
4. the HTLM model can also be used via the more traditional pretrain+finetune paradigm and achieve good performance on GLUE set.
5. the authors proposed a way to create the so-called auto-prompt, which can rival hand-crafted prompts (see additional questions below)

## Weakness and questions

1. the writing is inconsistent in several places and makes the related sections difficult to understand.
Sec 3.1 describes manual-NS and manual-S. But I can't find these settings in the experiments. in the experiments, HTLM-Manual is the only manual setting.
Also, related to writing, Table 2 font is really small and hard to read and is inconsistent with the rest of the paper.


2. The pretraining data, after cleaning is 23TB, which is a lot for retraining. So I looked up the amount of text data used in RoBERTa and BERT pretraining. According to https://arxiv.org/pdf/1907.11692.pdf , RoBERTa uses up to 0.16TB and BERT uses 0.013TB. So the proposed model is using more than 140x data than RoBERTa and more than 1,700x data than BERT. I wonder if there is any correlation between the amount of pretraining data vs the zero shot or finetune accuracy? what would happen to HTLM if it only had 0.013TB pretraining data instead of 23TB?

3. Fig 2 attempts to illustrate the idea of auto-prompting, which I'm a little confused about. In order the generate the template on the right, one has to get the pair of two paragraphs  and insert the four mask tokens around them. So in order to generate one prompt for the summarization task, the user has to prepare this one pair of sample. Is Auto-prompting zero-shot learning or in this case one-shot method? the authors titled section 4 as "zero/one-shot prompting" and the HTLM entry in Table 2 is under one-shot. but there is no further explanation if this is what the authors meant by one-shot.

4. Also in figure 2, after the HTML is generated on the right. Is there additional processing needed for using this prompt. Based on Figure 2. the model generates not only the HTML tags but also addition text: ` | The Washington Post`. so despite it being called auto-prompt, the user still has to verify if the model is generate 1)  syntacticly correct HTML tags and 2) if the model is generating additional/unwanted text. As I understand it, if the model generates this additional ` | The Washington Post` and the human user doesn't catch it, this text will be part of the prompt and be added to all samples during the testing , which presumably would be bad for performance. So now that it requires human verification of HTML syntactic correctness and any potential text hallucination, can we still call it "auto"?

5. What denoising pre training tasks are used exactly?
In intro and Figure 1, the authors mention title-tag denoising and class/id attribute denoising as examples for the model to pre-train with. Are there any other tasks the model is trained on? for example, there can be list and tabular elements in HTML. Does the model train with those structures?







**Summary Of The Paper:**

This paper presents HTML-based large scale language model pretraining. The model architecture and pre-training method are both based on BART. The author presents the resulting model, HTLM, as a strong zero-shot learner while being on par with other large transformer language models on the GLUE fine tune tasks. the authors show that via various benchmarks in their experiment section.

**Summary Of The Review:**

This paper presents HTML-based large scale language model pretraining. The model architecture and pre-training method are both based on BART. The author presents the resulting model, HTLM, as a strong zero-shot learner while being on par with other large transformer language models on the GLUE fine tune tasks.

While the authors present a lot of details in a linear fashion, there are still several gaps in the writing and would require clarification.

---

> ### Author Response · Authors · 2021-11-22
> **Re Reviewer 36N9**
>
> Thank you for your review. We'd like to address your questions and our paper's weaknesses.
>
> 1. In the new draft, we have cleared up the descriptions of the various types of prompting we did.
> 2. We used roughly half the amount of compute as previous BART models (in terms of batch size and the number of steps). We used this much data to ensure that we only touched every data point once. Bart, on the other hand, did dozens of iterations over all of its data. In general, multiple iterations over the data are okay for models such as BART, which only really generate ~15% of the document at a time. We just covered our bases by collecting more data, given the scalability of our pipelines.
> 3. We do auto-prompting by asking the model to recover the underlying document structure given any pieces of text. Another way to describe this is to ask the model to describe the coupling between pieces of text (i.e., document, summary) by generating document structure.
> 4. We also agree with you that it still requires some user interference to use the auto-prompt, specifically removing prefixes and suffixes; therefore, there's a bit of ambiguity between calling it zero-shot and one-shot.
> 5. We maintain the original denoising objective, which does random masking of spans over source sequences. Because random masking is the super-set of all masking policies, we see that our models can reason about a wide variety of HTML elements, including tables (which we showed in the NLG tasks). In the updated draft of the paper, we added sections of text to call this out explicitly.
>
> Thank you for your review. We hope you find our revisions satisfactory.

---

### Official Review · Reviewer_2QhH · 2021-10-30

**Correctness:** 3
**Technical Novelty And Significance:** 3
**Empirical Novelty And Significance:** 3
**Recommendation:** 6
**Confidence:** 4

**Main Review:**

Pros:
1. This paper studies a new direction in language model pretraining that goes beyond using plain text as the pretraining data. It effectively leverages the HTML data which can be obtained in large amounts via common crawl.
2. The paper is overall well-written with sufficient details and organized presentations.
3. The proposed model achieves superior performance on zero-shot summarization and does well on classification tasks.

Cons:
1. The technical contribution and novelty are rather weak. The model architecture and training objectives are largely based on BART, with some modifications to tailor for HTML-format training data.
2. The HTLM model is pretrained from a BART-large checkpoint which means it still needs to start from a language model that is trained on plain texts. It's unclear how necessary is this dependency (e.g. if HTLM is randomly initialized instead of continuing training from BART, will the results be much worse?)
3. Some presentations can be further improved. For example, it would be better to include an illustrative figure or explicit formulas showing an overview of how the BART pretraining objectives are tailored to the HTML-format training.
4. The full fine-tuning experiments in Sections 5 and 6 fail to cover recent state-of-the-art plain text pretrained models, like ELECTRA [2], DeBERTa [3] and COCO-LM [4]. And the statements that "HTLM improves over existing pre-training methods" and "hyper-text prompts provide more data efficiency to the HTLM model than plain text prompts do for existing LMs" seem to overclaim. Actually, the most recent plain text pretrained LM, COCO-LM, achieves better performance on GLUE than HTLM with fewer model parameters and has comparable performance with HTLM-R3F-Prompt which leverages additional fine-tuning techniques. I actually don't think HTLM (as the first PLM trained on large-scale HTML-format data) has to claim better performance than LMs trained on natural language texts on GLUE tasks to be impressive, but it is necessary to acknowledge that there exist better plain text pretrained LMs and give them credits.

Missing References:
[1] @inproceedings{devlin2019bert,
  title={BERT: Pre-training of Deep Bidirectional Transformers for Language Understanding},
  author={Devlin, Jacob and Chang, Ming-Wei and Lee, Kenton and Toutanova, Kristina},
  booktitle={NAACL-HLT},
  year={2019}
}
[2] @inproceedings{clark2020electra,
  title={ELECTRA: Pre-training Text Encoders as Discriminators Rather Than Generators},
  author={Clark, Kevin and Luong, Minh-Thang and Le, Quoc V and Manning, Christopher D},
  booktitle={ICLR},
  year={2020}
}
[3] @inproceedings{he2021deberta,
  title={DeBERTa: Decoding-enhanced bert with disentangled attention},
  author={He, Pengcheng and Liu, Xiaodong and Gao, Jianfeng and Chen, Weizhu},
  booktitle={ICLR},
  year={2021}
}
[4] @inproceedings{meng2021coco,
  title={COCO-LM: Correcting and contrasting text sequences for language model pretraining},
  author={Meng, Yu and Xiong, Chenyan and Bajaj, Payal and Tiwary, Saurabh and Bennett, Paul and Han, Jiawei and Song, Xia},
  booktitle={NeurIPS},
  year={2021}
}

Questions:
1. It is mentioned in the abstract and introduction that "element class and id attributes can encode categorical properties of documents". I'm not completely clear how these types of information are being leveraged in HTLM pretraining. Is the model explicitly trained to predict document class or ids?
2. HTLM has great performance on zero-shot summarization. Have you tried few-shot or even full training summarization?
3. The paper seems to mention some concrete examples/prompts are included in the Appendix, but I cannot find them.




**Summary Of The Paper:**

This paper introduces HTLM which is a language model pretrained on a large-scale web crawl hyper-text data. There are several contributions in the paper:
* A preprocessing step to filter out noisy components in the web pages is proposed. The resulting simplified format, Minimal-HTML (MHTML), is likely to be composed of high-quality documents which can be used for pretraining.
* A modified BART pretraining objective is proposed to inject noisy size hints to control the length of the span to be generated by the model during training.
* A new prompting method in the form of HTML templates is described to accomplish generation (e.g., summarization & table-to-text) and classification (e.g., GLUE) tasks.
* The resulting pretrained HTLM model has superior performance on zero-shot summarization and can do better than some existing language models pretrained on plain texts.

**Summary Of The Review:**

Overall, I appreciate the authors' efforts to propose the first PLM trained with HTML-format data and make it work. The pretrained model also shows impressive performance on some tasks (e.g., zero-shot summarization). However, I have some concerns about the paper (e.g., weak technical contribution and novelty, dependence on plain text PLMs, overclaims in the comparisons with state-of-the-art plain text PLMs, missing important references). If the authors could address my concerns in the rebuttal, I'm willing to adjust my rating.

---

> ### Author Response · Authors · 2021-11-22
> **Re: Reviewer 2QhH**
>
> Thank you for your detailed review! We’re glad you found our paper well written and appreciate our new direction in language model pre-training. We’d like to answer your questions first and then address the weakness mentioned in your review.
> 1. The model is not explicitly trained by masking certain HTML elements but rather depends on random masking. Therefore the model is capable of arbitrary forms of masking other than just masking element class/id attributes.
> 2. Masking out a title or headline is a very natural way to describe the summarization task whereas it’s a little harder to image an HTML prompt with multiple examples. The other downside to doing few-shot summarization is that it requires a significantly long sequence length since source documents are quite long, and sometimes summaries can be long as well. Furthermore we wanted the main story of the paper to be around prompting and not necessarily fine-tuning which is why all but one experiment focus on prompts (either evaluation or analytical experiments).
> 3. This was a mistake in the submission. We can fix this prior to the final draft.
>
> On the weaknesses of our paper.
> 1. We believe the novelty in our paper does not necessarily lie in the architecture but more in challenging the general trend of training on unstructured data. We argue that using the structure that naturally exists in documents allows us to prompt models in a completely new way. Furthermore, we believe the auto-prompting we introduced, whereby asking the model to recover document structure we naturally derive prompts, is a fundamental novelty as well.
> 2. We experimented with training from scratch and saw no fundamental need to have the start from BART. Given our compute budgets, we noticed that starting from BART would give us the lowest perplexities. On a follow-up project to this paper, we trained a 3B HTLM model from scratch that gave better perplexities than the 400M model from BART.
> 3. We have updated the draft of the paper to explain better how the masking was done.
> 4. We agree that the majority of novelty was not in the fine-tuning but rather in the prompting regime. The primary purpose of the fine-tuning was to show that training on structured documents does not necessitate poorer performance on purely natural language fine-tuning. The claim that “hyper-text prompts provide more data efficiency to the HTLM model than plain text prompts do for existing LMs'' comes directly from the “How much data is a prompt worth?” paper where the goal is to measure the direct contribution of a prompt and not necessarily the parameter-efficiency in the fine-tuning regime. We compared the standard models from that paper and added another baseline while also benchmarking HTLM. We’ve updated the claim to state we are competitive with other pre-training methods and not better. We will also include the references that you mentioned. We hope you find this satisfactory.
>
> Please let us know if you have any issues with our paper. Thank you once again for the detailed review.

---

> > ### Comment · Reviewer_2QhH · 2021-11-25
> > **Thanks for the response and update**
> >
> > I would like to thank the authors for the response and update. As mentioned in my original review, I appreciate the new direction of pretraining LMs on HTML-format data, as well as the empirical advantages of the HTLM model on several specific tasks/settings like generation and prompting. Also, my concerns about the paper were partly addressed by the update and response. Therefore, I have increased my rating to 6.
> >
> > While I understand that the rebuttal period was short and the authors may not have sufficient time to incorporate all desired changes (e.g., adding an overview figure and fixing the appendix to include some concrete examples) in the updated draft, I would hope to see them in the final version. In the meantime, I would like the authors to comment on the following points which still remain somewhat unclear to me:
> > 1. I'm still quite unclear on why the element class/id attribute information in the HTML data is beneficial for model training. Wouldn't their values vary a lot across different web pages and provide inconsistent/meaningless signals to the model (e.g., the same entity or class may be represented by completely different ids in different sources, and the ids may be random numerical/string values that barely encode semantic information)? Or are they always in the raw text format so that the element class/id attribute prediction is still essentially a word/token prediction task rather than a classification task? I hope the authors could clarify this in the next paper version.
> > 2. You mentioned that few-shot summarization would make the input sequence too long to be taken by the model; does it mean that HTLM is not able to leverage more training data in summarization tasks at all? Or it still can be trained on these data (in some way) but may not outperform plain-text LMs (like BART)? I buy the argument that HTLM is naturally fit to zero-shot summarization and I'm glad to see the impressive results, but I believe a more complete comment/explanation about the applicability of HTLM to different settings of text generation tasks would help the readers better understand the mechanism of HTLM.
> > 3. Regarding the dependency on a pretrained BART model, I'm still curious about the quantitative difference between continuing from BART and training from scratch given a fixed model size (it's not convincing to compare a 3B model to a 400M model). This study is important because this paper explores a new format (HTLM) of pretraining data, while the current prevalent pretraining data format is raw text. I imagine an important future direction to be how to use both of them because they obviously have their respective advantages. Continued training from a BART checkpoint seems to be an implicit way of leveraging both data types, but it makes it unclear to measure the respective benefit of either data type. I understand that it's costly to conduct the full training for this ablation, but it would be very helpful to show the difference between the two settings with a pilot study (e.g., trained on a subset of the entire corpus).

---

### Official Review · Reviewer_Pgkk · 2021-11-02

**Correctness:** 4
**Technical Novelty And Significance:** 4
**Empirical Novelty And Significance:** 4
**Recommendation:** 8
**Confidence:** 4

**Main Review:**

I really enjoyed reading this paper. The contribution is primarily empirical and is thorough. Modeling the HTML directly seems sensible and highly practical given the large amount of web data that can be used and all the organization that is already done in web-pages for applications like SEO. Things like the <title> tag seem to provide extra annotation to the text and the model seems to utilize these in summarization tasks. Overall I have nothing to complain about and think this is going to be a rich and interesting line of work - training LMs directly on structured text data.

I am curious if the authors performed any multi-lingual analysis on benchmarks like xtreme (https://arxiv.org/abs/2003.11080) - could be a potential future direction.

**Summary Of The Paper:**

This paper introduces a large-scale LM trained directly on the raw HTML in a large-scale web crawl (the common crawl corpus). The resulting model is able to utilize the structure in HTML documents for a variety of tasks such as zero shot summarization, fine-tuning, classification and more. It looks like using structured data for pre-training and creating prompts provides a variety of advantages in LMs. The paper has a comprehensive set of experiments and a thorough ablation study.

**Summary Of The Review:**

The paper presents a straightforward idea to pre-train large scale LMs directly on HTML. The empirical analysis is thorough, the results look very good, and the writing flows very nicely. Overall a very nice and fun read.

---

> ### Author Response · Authors · 2021-11-22
> **Re: Reviewer Pgkk**
>
> Thank you for your review! We are glad you enjoyed our paper! Our model was mainly trained on English documents; therefore, we don’t have too many expectations on being able to model multi-lingual data. But future projects should consider training jointly on a wide variety of languages. Thanks again for your review.

---

### Official Review · Reviewer_srNC · 2021-11-02

**Correctness:** 4
**Technical Novelty And Significance:** 3
**Empirical Novelty And Significance:** 2
**Recommendation:** 6
**Confidence:** 3

**Main Review:**

I would like to thank the authors for this intriguing piece of work – I particularly enjoyed reading the rigorous analyses of the proposed language model and am curious to see how the artifacts of this research will be leveraged by the broader NLP community. I would summarize the strengths and weaknesses of this paper as follows:

Strengths:
1. The idea of leveraging HTML text for pre-training large language models, and linking it to the structure of documents for supervision is quite interesting and creating. As the authors demonstrate via zero-shot summarization experiments, such a pre-training can be beneficial for specific NLP tasks like summarization.
2. The experiments in the paper are conducted on the GLUE benchmark. The analysis and experiments are rigorous and support the claims made in the paper. The paper is well-written and the contributions are easy to comprehend.
3. I particularly like the analysis that aims to understand the per-prompt-efficacy of HTLM and contrasts it with that of existing pre-trained language models.

Weaknesses:
1. The authors do not conduct their experiments on SUPERGLUE – a much more difficult benchmark than GLUE. This is in contrast to pre-trained language models like T5 that were evaluated on both GLUE and SUPERGLUE tasks. This seems to be an important evaluation that is currently missing from the proper.
2. It was hard for me to comprehend some of the choices made in the paper. For instance, the decision to skip MHTML documents whose ratio of text to HTML was not greater than 0.46 did not seem well justified to me. Perhaps explaining the filtering strategy in Section 2.1 would help. In some of the cases, it was not clear what the size hint was and how it was determined (e.g., in Table 1 for auto-prompting with size hints included).
3. Since the concept of size hint is one of the key contributions of this work, it would have helped to see how sensitive the results are with respect to various size hints (sensitivity of the downstream to +/-1 or 2 sizes). This analysis is currently missing from the paper.

Overall, I enjoyed reading the work. Since the contribution of this work hinges on rigorous evaluation, I would primarily encourage the authors to expand their evaluations to include the SUPERGLUE tasks, as some of the previous studies do.


**Summary Of The Paper:**

The paper proposes a new large language model, called HTLM, short for Hyper-Text Language Model. The language model has been trained on 25TB of HTML text. The authors argue that HTML tags provide valuable information regarding document-level structure. Besides the scale, the authors also adopt an optional pre-training strategy where it is possible to provide size hints to govern the length of the generated text or each mask (alongside the BART-like training objective). The trained language model achieves state-of-the-art performance on zero-shot summarization by prompting the model to predict the text within the <title> tags. The method also achieves state-of-the-art performance on several other benchmark NLP tasks, performing better than large pre-trained models that were only trained on text data (as opposed to hyper-text data). The authors also discuss manual and automated prompt construction, and their evaluation shows that the model can also perform better than competitive baselines on the multiple classification tasks. A follow-up analysis also shows that the per-prompt-efficacy is higher for HTLM than for text-only language models.

**Summary Of The Review:**

The work is technically sound, the core idea is creative, and involves a series of rigorous evaluations to substantiate the claims made. However, in line with some of the highly related previous works, I believe that the paper will benefit from evaluations on some additional NLP tasks that are considered more difficult (SUPERGLUE). Also, adding a few missing details and conducting a sensitivity analysis of the downstream performance with respect to the size hint would help the readers and practitioners who would use the pre-trained model for downstream tasks. I recommend a weak acceptance for the paper.

---

> ### Author Response · Authors · 2021-11-22
> **RE: Reviewer srNC**
>
> Thank you for your detailed review! We’re glad you found our paper intriguing and enjoyed our rigorous analysis of the language model.
>
> Our main goal of the paper was to show that leveraging structure in pre-training data leads to better zero-shot prompting rather than focusing on fine-tuning, so unfortunately we didn’t get much of a chance to focus on this aspect. That being said, we can add SuperGlue results to the final draft of our paper, given that the week of rebuttal was not long enough to run the SuperGlue benchmark.
>
> Furthermore, we agree that certain decisions made about the data processing could be explained more clearly and have updated the draft to discuss how we arrived at our choices. We have also added a short description of how the size-hints were selected in every case. We hope the changes suffice in mitigating Weakness 2 in your review.
>
> For Weakness 3, we agree that ablating over the size-hints would be an interesting addition. We’ll work on this and add it to the paper as well.
>
> Thank you for your detailed review! I hope you find our revisions satisfactory.

---

### Decision · Program_Chairs · 2022-01-20

**Decision:**

Accept (Poster)

**Comment:**

This paper develops a new large language model trained on 25TB of (simplified) HTML text data. The HTML tags provide valuable information about the document structure. The training adapted the BART denoising objectives (to inject noisy size hint to control generation length during training). The paper also studies various prompting methods for the model. The model achieves state-of-the-art performance on zero-shot summarization and several text classification tasks. Reviewers have found the motivation of pretraining with structured text convincing, and the results are good.